# On the Use of EBSD and Microhardness to Study the Microstructure Properties of Tungsten Samples Prepared by Selective Laser Melting

**DOI:** 10.3390/ma14051215

**Published:** 2021-03-04

**Authors:** Mirza Atif Abbas, Yan Anru, Zhi Yong Wang

**Affiliations:** Laser College of Engineering, Beijing University of Technology, Beijing 100124, China; yananru@bjut.edu.cn (Y.A.); zywang@bjut.edu.cn (Z.Y.W.)

**Keywords:** electron back scattering diffraction (EBSD), microhardness, plasma facing components (PFCs), additive manufacturing (AM), selective laser melting (SLM)

## Abstract

Additively manufactured tungsten and its alloys have been widely used for plasma facing components (PFCs) in future nuclear fusion reactors. Under the fusion process, PFCs experience a high-temperature exposure, which will ultimately affect the microstructural features, keeping in mind the importance of microstructures. In this study, microhardness and electron backscatter diffraction (EBSD) techniques were used to study the specimens. Vickers hardness method was used to study tungsten under different parameters. EBSD technique was used to study the microstructure and Kikuchi pattern of samples under different orientations. We mainly focused on selective laser melting (SLM) parameters and the effects of these parameters on the results of different techniques used to study the behavior of samples.

## 1. Introduction

Tungsten has been widely studied due to its unique characteristics and comportment under related fusion loading situations such as high melting point, low erosion rate, low sputtering yield, high thermal conductivity, and significant neutron irradiation resistance. Tungsten diverter has been proposed for the International Thermonuclear Experimental Reactor (ITER), and it is also a very important material for the Demonstration Power Plant Reactor (DEMO) [1]. Despite its importance, several efforts are required to explain and understand the problems affiliated with tungsten interactions, most importantly, during the maximum heat loads at which conditions are severe and will conclude the durability of the plasma facing components (PFCs). Obviously, strong heating exposure of the surfaces will ultimately cause dynamic recrystallization and grain growth, resulting in cracking. This is why it is so important for plasma-facing materials to consider the evolution of the microstructure according to ambient temperature [2,3,4].

In the coming years, nuclear fusion is assumed to have considerable promise in terms of electricity output within the world energy matrix. In addition to the fact that it does not produce greenhouse gas emissions, which are a major concern of the world’s primary source of energy, i.e., the burning of carbon fuels, nuclear fusion is inherently secure more efficient relative to traditional fission-based technology, and it does not impose threats of nuclear proliferation [5]. For a controlled and sustainable nuclear reactor, tungsten is the most suitable material for its chamber.

Electron backscattering diffraction (EBSD) provides us a very relevant data in a different orientation. It is a very powerful technique for quantitative microscopy. This method provides us data related to orientation and disorientation and can be extrapolated to represent strains, the extent of recovery, recrystallization, etc. In this technique, electron beams interact with the sample of tungsten and extract complementary information about texture, grain size, strain, and boundary characteristics. In the present paper, EBSD was used to study the grain microstructure of tungsten samples designed by different selective laser melting (SLM) parameters. Laser parameters used were scan speed, spacing, and laser power.

Microhardness is conducted to characterize the material’s resistance to plastic deformation. It is added to the material’s yield strength and is relatively non-destructive. Method for the measurement of mechanical properties is a surface indent in the order of 10 s performance in each dimension, by µm or less. Generally, a rigid intender is pressed into the sample and a force is used to make the indent is normalized by the size of the imprint to calculate hardness [6,7]. Vickers microhardness is frequently used to measure the hardness of additively manufactured materials because it allows high spatial resolution (measurements can be taken on the order of 100 µm up to 200 µm with an indent size of 50 µm) for finer measurements Nanoindentation can also be used. Nanoindentation is a great medium for the interpretation of the sub-micrometer volume mechanical reaction and thereby captures tiny shifts in the sub-surface microstructure that are not possible for traditional measures of hardness. Given the agreed sensitivity of the recrystallization Nanoindentation methods, the use of such measurements for non-destructive analysis should be practical.

In this paper, we are building on our recent research and further identifying the obstacles that hinder the efficacy of pure tungsten treatment through SLM. The ultimate purpose of the study was to use “method diagrams” as a basis to analyze the interactions between process variables and component output more systematically [8]. The observations were quantified in words, in addition to the microstructure and crystallographic texture, and the component density and nature of the defects made. The objective of the current study is to better understand the influence of grain size, hardness, texture, and grain morphology of additively manufactured tungsten [9].

## 2. Materials and Methods

The material used in the study was TEKMAT W-45 (Tekna, Suzhou, China) tungsten powder. The chemical composition of the powder is given in Table 1 below. The particle size distribution and surface morphology are shown in Figure 1a,b below. The crystalline metal powder has a high density and a more granular structure. These powders are suitable for additive manufacturing. Due to the spherical shape of the tungsten particle, it is suitable for the SLM process. The particle size distribution of tungsten particles varies from 8 microns to around 25 microns [10].

Unique parameters were selected for the manufacturing of specimens. The SLM equipment mainly included a Ytterbium fiber laser with a maximum power of 200 W, a laser beam of spot size of 40 µm, a high-speed laser scanner, a layering system, an atmosphere protection system, and computer-controlled software EOSPRINT Laser [11]. The amount of oxidant content inside the chamber was less than 0.10% to reduce the oxidation and balling phenomenon. The laser power, scanning speed, and hatching distance were set in the range of 110–170 W, 300–800 mm/s, and 0.03–0.08 mm respectively. The size of each sample was 5 mm × 5 mm × 5 mm. All the samples were manufactured on a stainless-steel substrate with a preheating temperature of 180–200 °C. This whole printing process lasted for approximately 7 h.

## 3. Results and Discussion

### 3.1. Microhardness

Seven Different samples were taken to examine the hardness of pure tungsten specimens fabricated by SLM. These results were later compared with the conventionally manufactured tungsten. During the whole procedure, each sample was placed on the Vickers hardness apparatus and five different points were taken on the specimen. For every specimen, the value of load was fixed and compression dwell time for each point was 10 s. With the help of the given formula, Vickers hardness (VH) was calculated. During the experiment, the value of load was 50 gf, which was approximately equal to 0.490 N applied [12]. The VH value was between 110 Kgf to 1400 Kgf. Laser parameters have also a very significant role in measuring the value of VH. The observations were recorded and the relationship between microhardness and other parameters were studied in the observation Table 2 given below.

In the observation Table 2, values of different parameters and average hardness have been presented. It is observed that these parameters have a direct impact on the value of Vickers hardness. The first sample was designed at a laser power of 130 watts with a scan speed of 500 mm/s and hatching space of 0.03 mm. When this sample was taken to measure the microhardness, five different points were taken on the sample and different values were taken, and finally, the average value was presented in the above-given Table 2. The average value of this sample was around 509 Kgf. When comparing this value with other samples that were designed at 150 laser power with a little difference of scan speed and hatching distance, it was found that the values of Vickers hardness were very close [13]. The values of the other two samples that were designed at the same laser power of 170 watts but with different scanning speed and hatch spacing indicate that in those two samples, the value of micro-hardness has been increased. The maximum value of hardness is around 762 Kgf.

It was observed that the mechanical properties of the SLM specimen were comparatively better than the conventionally manufactured tungsten. Referring to Table 3, when we compared the value of HV (microhardness) with other specimens of tungsten, it was found that the results are quite better as compared with the other manufacturing techniques. Similarly, when the ultimate compressive stress of SLM specimens was compared with other specimens, especially with the chemical vapor deposition (CVD) technique [14], it was realized that the laser melting technique is also good for ultimate tensile stress (UCS). In the SLM approach, high densification without massive cracks or pores will produce and this reasonable level of residual stress in laser melting parts may cause an increase in hardness and dislocation strength.

Figure 2 shows the relationship between different values of energy density and hardness of SLM-processed tungsten (W) at different energy densities microhardness has different values. The relationship between these quantities is not uniform.

One main reason for the variable relationship between these quantities is that when the load is applied to the sample, there is a significant collapse, which may cause low energy density. Another important reason is the overheating and load accumulation, which may increase the energy density which ultimately coarse microstructure and reduce microhardness.

Figure 3a shows the relationship between stress–strain curves at different energy densities at room temperature [22]. The relevant values of ultimate compressive strength are shown in Figure 3b.

Figure 3a represents the stress–strain curve of SLM-processed tungsten (W) under different strain energies, and Figure 3b is the bar chart of UCS vs. energy density [23]. The value of UCS is dependent on energy density. The value of UCS is crossing the value of 900 MPa at the value of 1000 J/mm^3^, but it was noticed that further increase in energy density may decrease the compressive stress. It has been observed that by increasing energy density the power particles may come closer and completely fuse together to form a dense part. It was noticed that further increase in energy density may cause visible phenomena such as grain boundaries and pores formation, which may consequently reduce the compressive strength.

### 3.2. Crystallographic Structure

To study the EBSD data, two samples were taken for examination; both the samples were designed at different SLM parameters. For diffraction, samples were prepared very carefully because the diffraction can only take place at the surface level, and it cannot work at a deeper level. During the diffraction, the electron can interact up to 100 μm on the surface of the sample. If there is deformation from 50 μm to 100 μm on the surface, then it is difficult to study the crystallographic structure. During the study, the sample was placed at a 70° angle, and the Kikuchi pattern was achieved on a phosphorous screen [23,24]. For each pattern, at least five different points were measured for accuracy of orientation. The electron beam is falling on a particular grain at different orientations of 100, 010 and 001. At the grain boundaries of the sample, the EBSD pattern is dark, but it is always bright in between the boundaries. After every 15°, grain boundaries change.

In Table 4, processing parameters were presented for the samples that were selected for electron backscattering diffraction. In the above-given samples, both samples were designed at the same hatch spacing but at different scan speeds and laser power [25]. For EBSD, the electron beam was accelerated at 15 KV at the acquisition frequency of 42.99 Hz, and the hit rate was approximately 80.81%.

Figure 4a shows of EBSD inverse pole figure (IPF) orientation map of pure tungsten specimen designed by SLM technique by using different parameters. The above-given tungsten sample was designed at a laser power of 150 watts with a scan speed of 300 mm/sec.

The above-given images are the EBSD IPF orientation of the second sample. This sample was designed at a scan speed of 400 mm/s and laser power of 110 watts, but the hatch spacing was the same for both samples. Both the samples have different manufacturing parameters; therefore, they have slightly different grain boundaries and grain structure. When we analyzed the maps in the vertical section, the formation of long columnar grain was evident along the build direction (BD) [26]. The re-growth structure is common in the melt pool process, where, in pure metals, supercooling is possible, which encourages nucleation in the melt ahead of the solidification front. The solidification further leads to grain growth. This solidification further leads to a mechanical property called anisotropy in metals. During the SLM, solidification occurs because of the deposited re-melted layer, which provides a substrate. For the formation of new grains, there is always a barrier because of nucleation energy, but there is no barrier for regrowth for preexisting grain, especially in pure metals. Solidification occurs continuously at preexisting grains present at the fusion boundary. The growth of grain is similar to conventional grain growth. The fastest growth is always in the direction parallel to the thermal gradient at the solidification front, which is normal to the rear melt-pool surface [27].

The grain size distribution of tungsten sample 1 and sample 2 is illustrated in Figure 5a–f, respectively, in which high angle boundaries (HAGBs) are highlighted by black lines. It is evident in Figure 5a–f that the first sample has higher angle grain boundaries compared to specimen 2. The main reason for the difference is SLM parameters is that specimen 2 is designed at great scan speed, and hence it has fewer high-angle grain boundaries. In the above-given figures, the grain diameter size-frequency distribution is also shown [28]. When we compared the grain diameter size–frequency distribution of sample 1 with the other sample, it was found that the majority of the grain size in the first sample is greater than 10 µm, as compared to the second sample, in which the majority has grain size less than 10 µm when examined in the build plane [29]. The columnar grain may also change by changing hatching distance and laser power. From the study, it was found that the melt-pool overlapping between the neighboring tracks may lead to greater grain size diameter [30].

Pole Figure 6a,b shows the surfaces taken from the cross section view of EBSD maps of samples 1 and sample 2. In the pole figure diffraction beam falling on a particular grain at orientations of 100, 110 and 111 [31]. The pole figures of both samples have different textures. Sample 1 has a strong orientation aligned with Y. There is fairly a uniform scattering of the pole around this alignment. The surface can be described as the best fiber texture along 110. The second sample has a different orientation compared to the first sample. In Body Centered Cubic (BCC) metals, preferential growth direction during solidification is in <100> direction but here it is in <110> direction [32]. To study the texture further more investigation is required. Another growth direction was noted during the manufacturing in response to parameter changes. Different manufacturing techniques produce variable orientation; for example, Directed Energy Deposition (DED) system produces <110>//Z orientation, while in AM, <001>//Z texture is commonly observed where the melt is shallow and elongated because of high scanning speed and low thermal conductivity (i.e., titanium) [32]. Additive SLM manufacturing is a comparatively better process because, in this process, we have a deeper and bowl-shaped melt pool, and this approach increases the angle between the crystal growth and Z plane. Materials such as tungsten are more likely to produce inclined solidification along the Z plane because of their thermal conductivity and surface tension, as compared to other materials.

## 4. Conclusions

This research has recognized certain of the key hurdles that boundary the usefulness of SLM treatment of pure tungsten. The influence of key handling factors has been validated by creating a process window in which trial models could be built with reasonable optical quality, with varied laser power and scanning speeds. It has been observed that the values of SLM parameters have great impact manufacturing of different tungsten samples. When measuring the microhardness, it is observed that AM specimens have comparatively better value compared to other samples, which were fabricated by different conventional techniques. In further investigation related to EBSD characteristics of tungsten specimens, texture and grain size were reasonably small compared to other designed tungsten models. The consideration of part reliability, in terms of perviousness, cracks and microhardness presented that our approach was capable to liquefy the pure W powder and that a change in parameters was complemented by a rise in comparative values such as microhardness and energy density. Few samples were pre-disposed to cracking, but comparative densities were achieved up to 98% by variable parameters. The quality of the W sample was adequately good for the use of medical radioactivity shielding and atomic imaging. The SLM samples are more accurate and complex structure compared to other ordinary samples. During the microstructure investigation, it was revealed further that these samples generated columnar grain configuration by an epitaxial growth procedure. Pure metals show unusual grain structures under different manufacturing parameters. This unusual behavior may be related to the deeper melt-pool shape that normally occurs in the SLM technique, which is because of high thermal conductivity, surface tension, and 67-degree raster rotation direction between deposited layers. The main features of this research have shown that through this technique’s very high-quality components can be manufactured comparatively with low cost, complex structure, and small shapes.

## Figures and Tables

**Figure 1 materials-14-01215-f001:**
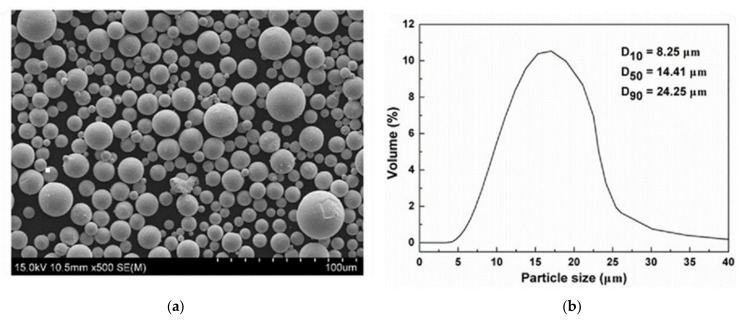
(**a**) shows the SEM morphology of tungsten (W) and (**b**) shows the particle size distribution.

**Figure 2 materials-14-01215-f002:**
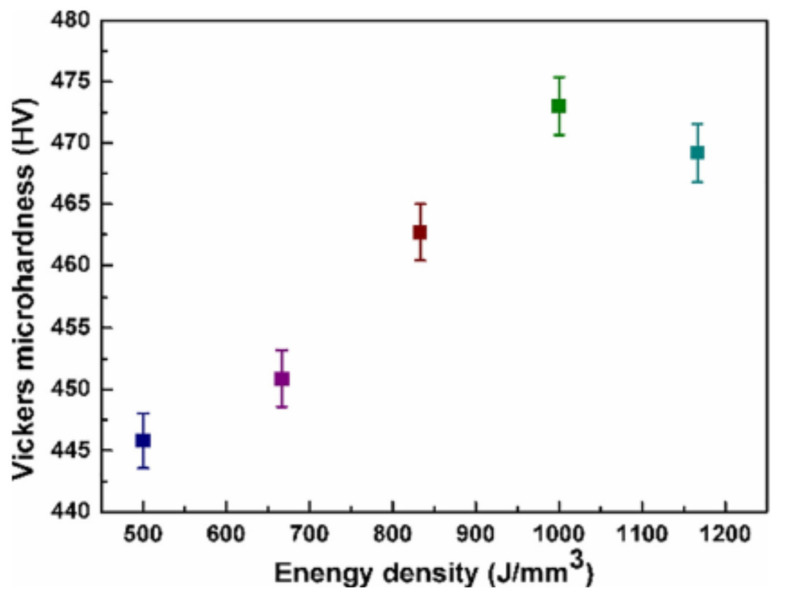
The relationship between energy density and microhardness of pure tungsten (W) manufactured by the selective laser melting (SLM) technique.

**Figure 3 materials-14-01215-f003:**
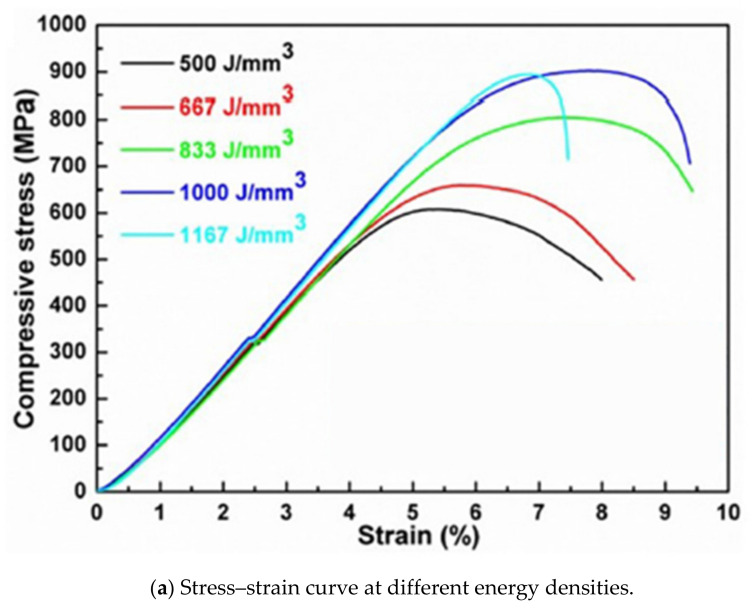
(**a**) stress–strain curve at different energy densities and (**b**) ultimate tensile stress (UCS) vs. energy density.

**Figure 4 materials-14-01215-f004:**
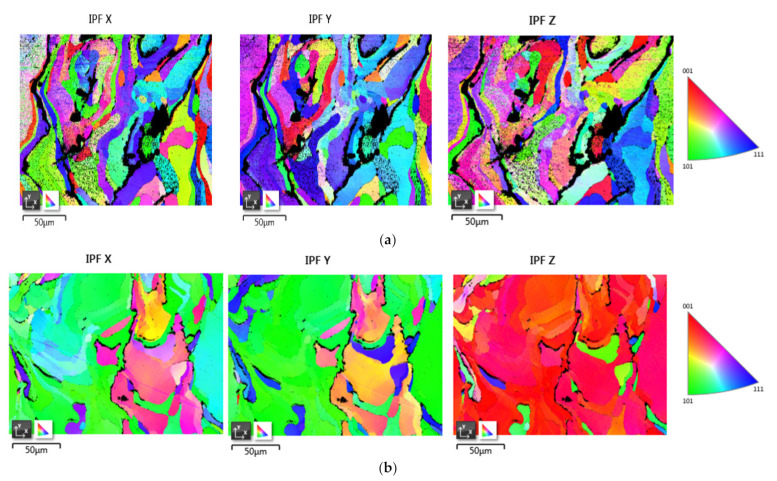
(**a**) EBSD inverse pole figure (IPF) orientation map of sample one is shown in figures. The IPF coloring is the plane’s normal direction parallel to X, Y, and Z direction. (**b**) EBSD inverse pole figure (IPF) orientation map of sample 2 is shown in figures. The IPF coloring is the plane’s normal direction parallel to X, Y, and Z direction.

**Figure 5 materials-14-01215-f005:**
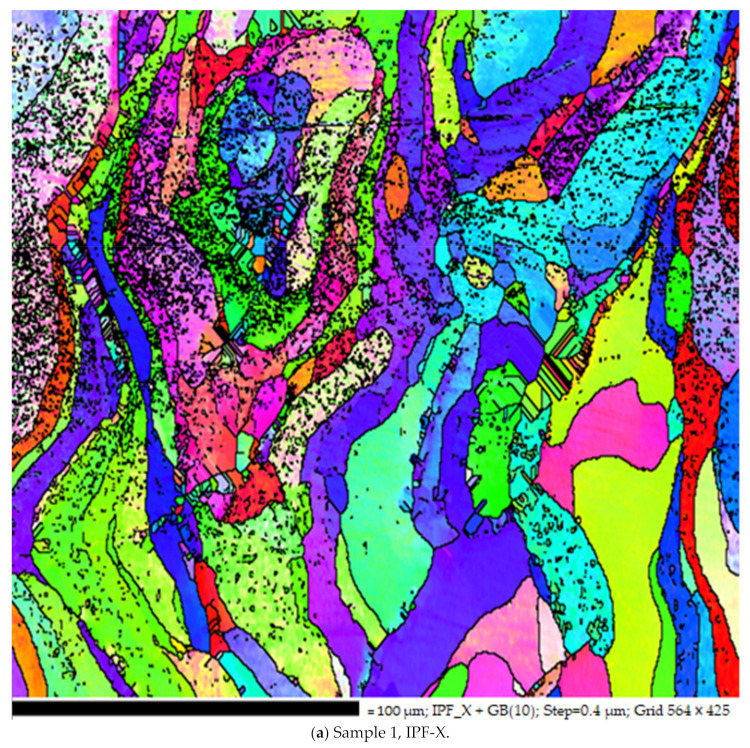
(**a**–**c**) describe the electron backscatter diffraction showing black lines as high angle grain boundaries >15° and grain size distribution in the plane measured for sample 1. (**d**–**f**) describe electron backscatter diffraction showing black lines as high angle grain boundaries (HAGBs) >15° and grain size distribution in the plane measured for sample 2.

**Figure 6 materials-14-01215-f006:**
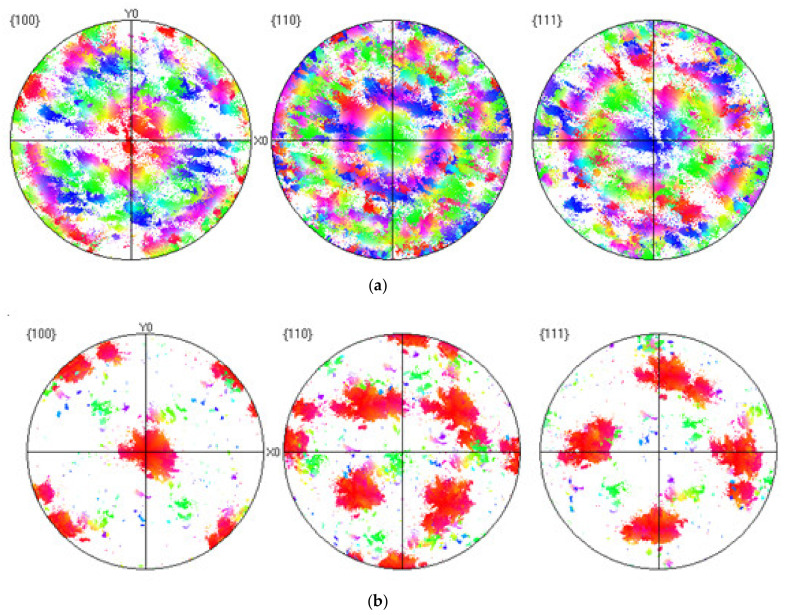
(**a**) Pole figures depicting the tungsten textures measured in the planes for sample 1. (**b**) Pole figures depicting the tungsten textures measured in the planes for sample 2.

**Table 1 materials-14-01215-t001:** Chemical composition of tungsten powder (wt. %).

Element	Tungsten (W)	Oxygen (O)	Carbon (C)	Sulphur (S)
wt. %	Balance	0.0016	0.0063	0.0022

**Table 2 materials-14-01215-t002:** Observation table for microhardness.

No. of OBS	Manufacturing Parameters			Dwell Time (s)	HV (Kgf)	HV (GPa)
	Laser Power (watt)	Scan Speed (mm/s)	Hatch Spacing (mm)		Average Value	
1	130	500	0.03	10	506.9	4.97
2	150	500	0.03	10	438.0	4.23
3	150	500	0.04	10	458.7	4.45
4	150	400	0.08	10	451.7	4.42
5	150	500	0.08	10	436.2	4.27
6	170	500	0.03	10	762.7	7.47
7	170	400	0.04	10	484.5	4.75

**Table 3 materials-14-01215-t003:** The relationship between mechanical properties and manufacturing techniques.

Manufacturing Technique	HV	Ultimate Compressive Stress (MPa)
SLM (selective laser melting)	450–480	900–920 (This study)
CVD (Chemical vapor deposition)	410–430	770–790
PM (power metallurgy)	430–450	1000–1150 ([15,16,17])
SPS (spark plasma sintering)	290–310	960–990 ([18,19,20,21]

**Table 4 materials-14-01215-t004:** SLM parameters for samples selected for electron backscatter diffraction (EBSD).

No. of Observations	Scanning Speed (mm/s)	Laser Power (watt)	Hatch Spacing (mm)
Sample 1	300	150	0.08
Sample 2	400	110	0.08

## Data Availability

Data sharing is not applicable to this article.

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
