# Peer review of "On the Use of EBSD and Microhardness to Study the Microstructure Properties of Tungsten Samples Prepared by Selective Laser Melting"

_materials, 2021, doi:10.3390/ma14051215_

Round 1
Reviewer 1 Report
The article is relevant, since it is aimed at studying the hardness of tungsten at various parameters. Among the shortcomings, it is necessary to single out the unrepresentative design of the graphs (Figure 2, b, Figure 4, a, b). It is impossible to see the contents on the graphs (Figure 4 a, b) - it is necessary either to present them in the form of a separate (larger) figure, or to delete them.
Author Response
Dear respected Reviewer please look at the attached file thanks

Reviewer 2 Report
This study investigates into the use of electron back scatter diffraction (EBSD) and micro-hardness to obtain the microstructural properties of the specimens made of tungsten. This is an interesting report. However I recommend some major amendments before the paper can be recommended for publication.
1) The authors need to explain what has really been achieved in the abstract. It isn't currently informative enough. 2) The entire experimental apparatus and test procedures must be provided and discussed by the others. 3) Where did you obtain the electron backscatter diffraction parameters from? 4) Is there any particular reason as to why the VH values were selected to be within the range of 110 to 1400? Please clarify! 5) It's not clear at all how the ultimate compressive strength of tungsten was obtained? Have you obtained it experimentally? If yes, what have the procedures been? How many sample sizes have been used? Has calibration been required? None of this has been even mentioned in the manuscript, much less discussed. The same goes for energy density. 6) Fig.4c is not legible. It must be replaced. 7) The figure under Fig.4 doesn't have any caption. It's not clear what it shows. 8) extensive language and typo revision is required.
Author Response
Dear Respected Reviewer please look at the attached file thanks.

Reviewer 3 Report
In this study, the authors show the results of mechanical properties and microstructure of several Tungsten Samples. However, there are several general concerns about it that should be review:
- English style and grammar. There are many grammar mistakes concerning capitals after dots, not adequate expressions, etc. Also, the format of text is not regular, with high initial spaces in some lines and very small in others, tables parted in two pages and so on.
- Description of experimental methods: Table 2 and Figure 2.a show Compressive stress results, but it is not described how they were obtained.
- In point 2.2 it is said that “samples were prepared very carefully”. This is not technically correct, please describe the preparation method in detail.
- It will be useful to include the optical or electronical micrograph of the samples, not only EBSD images.
- Why do you only choose two samples for EBSD study? What about the others? It seems that the study is not still finished.
Author Response

(The authors gave the same response as above.)

Reviewer 4 Report
Dear authors,
The results presented in the manuscript entitled “On the use of EBSD and micro-hardness to study the micro-structure properties of tungsten samples prepared by selective laser melting” do not represent the required level from the scientific point of view and this is the reason why I must recommend its rejection.
In the manuscript the Materials and methods section is completely omitted. The section Experimental should be renamed to Results and Discussion. However, discussion is the main weakness of this manuscript. The manuscript looks like a report, not scientific paper. There is too much academic information about used methods. These knowledge is widely available in the open literature and it is not required to write how the Vickers hardness is measure or how the EBSD is doing.
It is not clear from the manuscript who is the author of the results presented in Table 2. They are probably not authors, so there are no citations of the works from which the results were taken. However, if these results are obtained by the authors, there is a lack of information about detailed processing routes and obtained other properties, e.g. density. That is why it is very hard to compare this results and make some conclusions.
The authors presented in Table 1 the Vickers hardness of obtained samples in GP unit. I think it is a big error, and hardness should be presented in kgf unit. Hardness of 763 GPa is more or less 7 times higher than hardness of diamond. It means that the authors has completely do not understand what they investigate.
The IPF legend is not visible in Figure 3. The resolution is to low also in Figure 4. From these micrographs it can not be analyzed the ratio of LAGB and HAGB.
In general, this manuscript does not bring any new knowledge to materials science.
Author Response

(The authors gave the same response as above.)

Round 2
Reviewer 2 Report
The authors have satisfactorily taken my comments into account and revised the manuscript accordingly. In light of such improvements, I'd like to proceed and accept this version of manuscript for publication in Materials.
Author Response
we are thankful to the respected reviewer for giving us a very positive feedback on the resubmission of our manuscript.

Reviewer 3 Report
The authors have satisfactorily improve the manuscript according the comments.
Author Response
we are thankful for the kind support and encouragement of our respected reviewer.

Reviewer 4 Report
Dear Authors,
Thank you for taking into account the reviewer’s suggestions bulleted in the first review. However, the manuscript is not improved significantly and in the present form it cannot be published in the Materials journal. I cannot found the Materials and methods section. The Materials and methods and Results and discussion sections should be clearly divided. The figures, despite the fact that are bigger in size, are characterized by still the low resolution. I suggest to present a more complementary work and resubmit the manuscript one more time.
Author Response
We are thankful for a very positive feedback of our respected reviewer.
